# Recent Advances in Understanding the Mechanisms of Elemene in Reversing Drug Resistance in Tumor Cells: A Review

**DOI:** 10.3390/molecules26195792

**Published:** 2021-09-24

**Authors:** Tiantian Tan, Jie Li, Ruhua Luo, Rongrong Wang, Liyan Yin, Mengmeng Liu, Yiying Zeng, Zhaowu Zeng, Tian Xie

**Affiliations:** 1School of Pharmacy, Hangzhou Normal University, Hangzhou 311121, China; 18438612030@163.com (T.T.); jie.li@stu.cdutcm.edu.cn (J.L.); 18327602598@163.com (R.L.); rong123450609@163.com (R.W.); liyanyin18@163.com (L.Y.); 15275766469@163.com (M.L.); 2Key Laboratory of Element Class Anti-Cancer Chinese Medicines, Engineering Laboratory of Development and Application of Traditional Chinese Medicines, Collaborative Innovation Center of Traditional Chinese Medicines of Zhejiang Province, Hangzhou Normal University, Hangzhou 311121, China; 3College of Pharmacy, Chengdu University of Traditional Chinese Medicine, Chengdu 611137, China; 4School of Traditional Chinese Medicine, Guangdong Pharmaceutical University, Guangzhou 510006, China

**Keywords:** elemene, multidrug resistance, mechanism, tumor

## Abstract

Malignant tumors are life-threatening, and chemotherapy is one of the common treatment methods. However, there are often many factors that contribute to the failure of chemotherapy. The multidrug resistance of cancer cells during chemotherapy has been reported, since tumor cells’ sensitivity decreases over time. To overcome these problems, extensive studies have been conducted to reverse drug resistance in tumor cells. Elemene, an extract of the natural drug *Curcuma wenyujin*, has been found to reverse drug resistance and sensitize cancer cells to chemotherapy. Mechanisms by which elemene reverses tumor resistance include inhibiting the efflux of ATP binding cassette subfamily B member 1(ABCB1) transporter, reducing the transmission of exosomes, inducing apoptosis and autophagy, regulating the expression of key genes and proteins in various signaling pathways, blocking the cell cycle, inhibiting stemness, epithelial–mesenchymal transition, and so on. In this paper, the mechanisms of elemene’s reversal of drug resistance are comprehensively reviewed.

## 1. Introduction

Cancer is a fatal disease and a global public health problem due to its high mortality rate [1]. Chemotherapy is one of the most common treatments for cancer. In recent years, the resistance of chemotherapy to anticancer drugs has become a major problem in oncology, leading to the failure of anti-tumor therapy mainly due to drug efflux and metabolism, tumor cell heterogeneity, genetic or epigenetic changes in cancer cells induced by tumor microenvironment stress, etc. Some cells evade death in each round of chemotherapy, and acquired drug resistance is developed in tumor cells after several rounds of standard drug treatment. To overcome multidrug resistance in tumor cells, a combination of chemotherapeutics and reversal agents has been extensively studied [2]. Various reversal agents have been developed, but most have failed in clinical trials due to severe adverse reactions. Therefore, since drug resistance and cross-resistance can occur after chemotherapy, researchers have focused on natural drugs with multiple targets. Many natural products of Chinese herbal medicine are reported to have anti-proliferative and anti-angiogenesis effects, promote tumor cell apoptosis, and inhibit tumor metastasis. In addition, these natural products can also regulate autophagy, reverse multidrug resistance, balance immunity, enhance chemotherapy in both in vivo and in vitro studies, etc. [3]. Recently, several natural products have been reported to enhance sensitivity or overcome resistance to anticancer drugs, including elemene, curcumin, Shenqi Fuzheng Injection, etc. [4].

Chinese medicine *Curcuma zedoaria* and Chinese medicine *Curcumae radix* are derived from different parts of the same plant—*Curcuma wenyujin*. *Curcuma zedoaria* originates from the rhizoma of *Curcuma wenyujin*, while *Curcumae radix* is from the tuberous root of *Curcuma wenyujin* (Figure 1). *Curcuma wenyujin* belongs to the family *Zingiberaceae* and the genus *Curcuma* [5]. Although *Curcuma zedoaria* and *Curcumae radix* are well recognized in Chinese medicine from past decades, their constituents such as elemene were not identified until later [6]. Elemenes (1-methyl-1-vinyl-2,4-diisopropenyl-cyclohexane) are among the 109 sesquiterpenes present in turmeric [7]. Elemene is a sesquiterpene compound with anticancer activity, extracted from traditional Chinese medicine *Curcuma zedoaria* and developed in China [8]. The chemical name is 1-methyl-1-vinyl-2,4-2 isopropenyl-cyclohexane, formula C_15_H_24_, relative molecular mass (Mr) 204, it has a small volume, and it is lipotropic to cross the blood–brain barrier [9]. The main component of elemene is β-elemene, whose chemical name is (1S, 2S, 4R)-1-vinyl1-methyl-2-4-double(prop-1-en-2YL) cyclohexane. The components of elemene have been analyzed by Gas Chromatography–Mass Spectrometry (GC–MS) and identified to contain β-elemene, γ-elemene, and δ-elemene (Figure 2) [7,10]. A variety of chemical constituents of elemene were further identified by Davidson et al. [11]. Elemene crude drug extracted from *Curcuma zedoaria* is a mixture consisting of β-elemene, γ-elemene, and δ-elemene. β-elemene is the major component that is the most active against cancer, and γ-elemene and δ-elemene are auxiliary components that can synergistically enhance the anticancer effect of β-elemene. β-elemene is a naturally occurring compound isolated from *Rhizoma zedoariae*. δ-elemene is also a terpene, and a structural isomer of β-elemene with three consecutive isoprene units. Some studies show that isomers of elemene exhibit anticancer activities in different cancer cell lines. The molecular compatibility theory identifies these three compounds as the main components used in the treatment of tumors. In the clinic, elemene injectable emulsion has been approved by CFDA (national medicine permission number H10960114) for the treatment of gliomas and brain metastases, and elemene oral emulsion has been approved by CFDA (national medicine permission number H20010338) for the adjuvant treatment of esophageal and gastric cancer with symptom improvement [8]. Elemene injectable emulsion as an antitumor drug was used in combination with chemoradiotherapy conventional regimens for lung cancer, liver cancer, esophageal cancer, nasopharyngeal cancer, brain tumors, bone metastatic cancer, and other malignant tumors, which can enhance the efficacy and reduce the toxic side effects of chemoradiotherapy. The main components of elemene injectable emulsion and elemene oral emulsion are all β-elemene, γ-elemene, and δ-elemene [8]. A review of the research progress on the antitumor efficacy and clinical application of elemene over the past 20 years by Shi et al. indicates that elemene has antitumor effects in the clinical field of liver cancer, lung cancer, gastric cancer, esophageal cancer, bladder cancer, colorectal cancer, pancreatic cancer, leukemia, multiple myeloma, head and neck nasopharyngeal cancer, and related metastatic cancers. Additionally, elemene can improve patient quality of life and is a drug that is safe to use with few adverse effects [12].

In recent years, elemene anti-tumor clinical research has been accumulated and deepened, and relevant clinical meta-analysis has been presented. In the clinic, elemenum emulsion was initially used to treat malignant effusions, and a prospective multicenter phase III clinical trial demonstrated that, under elemenum emulsion treatment, the response rate was 77.6% in patients with malignant pleural effusion and 66.1% in patients with abdominal effusion, with no major adverse effects [13]. Subsequently, it has been found clinically that elemene adjuvant chemotherapy has a synergistic effect. In a clinical trial, β-elemene combined with chemoradiotherapy concurrent therapy prolonged overall survival and progression free survival and increased 3-year overall survival in 102 patients with esophageal cancer [14]. In a clinical trial of combination therapy of β-elemene and temozolomide (TMZ) for glioblastoma, compared with the control, patients who received β-elemene showed significantly longer median progression-free survival (PFS) (8 months vs. 11 months; *p* < 0.001) and overall survival (OS) (18 months vs. 21 months; *p* < 0.001) [15]. In a meta-analysis, the clinical efficacy of elemene in the treatment of malignant pleural effusion in lung cancer was superior to that of cisplatin (DDP) [16]. In addition, elemene injectable emulsion has an adjuvant chemotherapy effect, which may improve the efficacy of chemotherapy, improve the quality of life, and reduce the adverse effects of platinum-containing chemotherapy drugs [17]. In a meta-analysis, elemene injectable emulsion was used as an adjuvant treatment to platinum-based chemotherapy in patients with stage III/IV non-small cell lung cancer, and 15 randomized clinical trials (RCTs) recruiting 1,410 patients with stage III/IV NSCLC were included. Compared with platinum-based chemotherapy (PBC) alone, the combination of elemene injectable emulsion and PBC can improve the disease control rate (DCR), objective response rate (ORR), 1- and 2-year survival rates, quality of life (QOL), CD4^+^T cell counts, and the CD4^+^/CD8^+^ratio and can reduce severe toxicities by 58% [18]. Sixteen RCTs matched the selection criteria, which reported on 969 subjects. Compared to chemotherapy alone, elemene combined with chemotherapy in the treatment of gastric cancer (GC) may increase the efficiency of ORR, improve their life quality based on Karnofsky performance status (KPS), and reduce adverse reactions, including leukopenia, neutropenia, anemia, and thrombocytopenia [19]. In unresectable hepatocellular carcinoma (HCC) in a total of 10 studies involving 543 patients, compared to transcatheter arterial chemoembolization (TACE) alone, TACE + elemene injectable emulsion may improve the ORR and the 1-year survival rate for HCC patients. The results showed that the ORR was significantly improved in the combined treatment group compared to the TACE alone group, and TACE + elemene significantly increased the 1-year survival rate. We also found no significant difference in gastrointestinal reactions, fever, or bone marrow suppression between the two groups [20]. These data suggest that β-elemene has a synergistic effect in the treatment of malignant diseases in combination with chemotherapy, and improves the prognosis.

In addition, for the phenomenon of multidrug resistance in cancer, elemene has been shown to play a role in reversing drug resistance in some drug-resistant cancers. In a cell-derived xenograft (CDX) tumor model with gastric cancer resistant cell line SGC-7901/Adr, the elemene treatment group had an inhibitory effect on tumor growth compared with the control group [21]. In a clinical observation, a total of 42 patients with advanced gastric cancer were selected and compared between elemene emulsion combined with chemotherapy and single treatment. The results showed that the response rate was 9% in the elemene emulsion alone treatment group, 0% in the chemotherapy alone group, and 28.6% in the elemene emulsion combined chemotherapy group. These data suggest that elemene emulsion combined with chemotherapy increases the sensitivity of tumor cells to chemotherapeutic drugs and reverses the resistance to chemotherapeutic drugs in patients with advanced gastric cancer [22]. Its mechanism of action has attracted many people’s interest to study its molecular mechanism and target [22].

Elemene is widely used in the treatment of malignant tumors such as lung cancer, gastric cancer, and breast cancer, where it regulates a variety of molecular signaling pathways and plays anti-tumor roles with fewer adverse effects [23]. β-elemene has anti-tumor effects, including playing a role in a variety of signal transduction pathways [24], inducing tumor cell apoptosis, blocking the tumor cell cycle, and reversing chemotherapy drug resistance [25]. Previous reports demonstrate that elemene has multitarget antitumor activity, triggers apoptosis through the activation of intrinsic pathways, and triggers cell cycle arrest by activating the p38 mitogen-activated protein kinase (MAPK) pathway [5]. Recent studies have found that β-elemene can alter inflammation and tumorigenesis, and its mechanism of action is crucial to macrophage infiltration and M2 polarization, which in turn regulates immune disorders. In addition, β-elemene not only regulates different inflammatory factors such as Tumor Necrosis Factor Alpha (TNF-α), Interferon (IFN), transforming growth factor Beta (TGF-β) and interleukin-6/10 (IL-6/10), but also regulates oxidative stress, has excellent anti-inflammatory and anti-tumor effects, and can alter the inflammatory microenvironment of tumors [26].

In recent years, elemene has been reported to play a role in anti-drug-resistance; however, the mechanism of action has not been fully understood. Some studies have demonstrated that the reversal mechanism of β-elemene may be related to the blocking of the efflux function of P-glycoprotein (P-gp), a reduction in the level of P-gp protein [27], the induction of apoptosis or cell cycle arrest, and a reduction in the activity of Nuclear Factor Kappa β (NF-κβ) signaling. Currently, in vitro and in vivo studies of β-elemene suggest that it can be used to treat certain multidrug resistance (MDR) cancers [5]. This review discusses in detail several reported mechanisms of elemene resistance. In addition, the mechanisms of elemene’s reversal of drug resistance are reviewed to provide ideas and references for subsequent research and clinical strategy in elemene’s reversal of drug resistance.

## 2. Mechanism of Multidrug Resistance in Tumor Cells

Chemotherapy is the most commonly used treatment strategy in clinical cancer treatment. Chemotherapy drugs inhibit the growth and metastasis of tumors to a certain extent, but with the long-term use of chemotherapy drugs, tumor cells produce MDR, resulting in chemotherapy failure [28,29]. MDR has been reported in a variety of cancers, including lung cancer, stomach cancer, leukemia, breast cancer, squamous cell carcinoma, colorectal cancer, and brain glioma, etc. [30]. The MDR phenotype is characterized by cross-resistance to a series of anticancer drugs with distinct structures and mechanisms of action [31].

The molecular mechanism of malignant tumor drug resistance is very complex and elusive. Drug resistance can be divided into two types: intrinsic resistance and extrinsic resistance [29]. MDR can either be pre-existing or induced by anticancer drugs. Intrinsic resistance is a pre-existing resistance, arising from genetic and epigenetic alterations in cancer cells [30]. Extrinsic resistance is also known as acquired resistance. In most cases, the mechanisms of acquired resistance are divided into three distinct processes: clonal selection, the activation of signaling pathways, and the triggering of resistance mechanisms. These three processes usually occur in parallel and are closely related to each other. Clonal selection refers to treatment that can lead to the death of a large number of drug-sensitive cells and subsequent proliferation of pre-existing resistant cells. The activation of signaling pathways refers to the temporary or permanent activation of various signaling pathways within tumor cells, enabling them to survive and develop acquired resistance during treatment. The triggers of resistance mechanisms are extracellular signals released by dying tumor cells, which trigger the formation of various resistance mechanisms in adjacent cells [32].

MDR in cancer has been investigated extensively since the discovery of a drug efflux pump, P-gp, in 1976 [33]. Multidrug resistance arises via many unrelated mechanisms, such as the overexpression of energy-dependent efflux proteins, a decrease in the uptake of the agents, an increase or alteration in drug targets, the modification of cell cycle checkpoints, the inactivation of the agents, the compartmentalization of the agents, the inhibition of apoptosis, and aberrant bioactive sphingolipid metabolism [34]. Chemotherapy agents can be involved in the selection of cancer stem cells, resulting in elevated drug resistance and enhanced tumorigenicity [35]. Molecular pathways are involved in the microRNA-mediated regulation of multidrug resistance [36]. Abnormal expression of ATP-binding cassette (ABC) transporter can increase the efflux of drugs from cancer cells to decrease the intracellular drug concentration, thus decreasing the efficacy of chemotherapy drugs, the inhibition of apoptosis, abnormal activation of the signaling pathways, and tumor drug resistance.

## 3. Mechanism of Elemene Reversing Multidrug Resistance in Tumors

The study of reversal agents of chemotherapy has become a hot topic in chemotherapy research. Many studies have confirmed that natural drug β-elemene can reverse the multidrug resistance of breast cancer [37], ovarian cancer [38,39], non-small cell lung cancer (NSCLC) [40], leukemia [41,42,43,44], human osteosarcoma [45,46,47], glioma [48], hepatocarcinoma [49,50], gastric cancer [51,52], etc., to chemotherapy.

In recent years, many mechanisms of elemene’s reversal of tumor multidrug resistance have been reported, and mainly include: (1) inhibiting the efflux of ABCB1 transporters (breast cancer resistance protein (BCRP) and P-gp) [27,53,54], (2) reducing exosome delivery [55,56], (3 inhibiting the EGFR-ERK/AKT pathway by downregulating the expression of miR-1323 [57], (4) inhibiting the PI3K–AKT signaling pathway by increasing Phosphatase and Tensin Homolog (*PTEN*) gene expression, (5) through the mitochondrial pathway of apoptosis [58,59], the upregulation of the Caspase-3 protein, a decrease in the expression of resistance genes multidrug resistance gene1 (*MDR1*), Lung Resistance protein (*LRP*), and tumor suppressor (*TS*) [60], the inhibition of the Signal Transducer and Activator of Transcription (STAT3) signaling pathway, the inhibition of the Cyclin-Dependent Kinase 8 (*CDK8*) gene and the expression of the *P21* gene to induce tumor cell apoptosis resistance [61,62], (6) the regulation of the cell cycle [63], (7) inducing autophagy [64,65], (8) inhibiting stemness [66,67,68], (9) inhibiting epithelial–mesenchymal transformation (EMT) [69], and (10) increasing the expression of estrogen receptor α (ERα) [70], etc. Tumor resistance can also be reversed by inhibiting the c-Met signaling pathway to inhibit P-gp function [71], upregulating E3 ubiquitin ligase, c-Cbl (Casitas B-lineage lymphoma, Cbl), and Cbl-b [72], inhibiting the ERK1/2-Bcl-2/Survivin pathway [73], inhibiting the PI3K/Akt/mTOR signaling pathway, and preventing the degradation of copper transporter 1(CTR1) [74]. Detailed mechanisms of elemene reversing multidrug resistance in tumor cells are shown in Figure 3, Figure 4 and Table 1.

### 3.1. Inhibition of ABCB1 Transporter Efflux (BCRP and P-gp)

Active efflux of drugs through the ABC transporters is a common mechanism of drug resistance. A total of 48 ABC transporters have been found, among which P-gp, multidrug resistance-related protein (MRP), and BCRP can confer MDR [82]. P-gp also MDR1/ABCB1, multidrug resistance-associated protein 1 (MRP1/ABCC1), MRP2 and BCRP/ATP-binding cassette subfamily G member 2 (ABCG2) are considered to be the main factors inducing MDR [83,84]. When the ABC transporters are overexpressed in cancer cells, they can confer cross-resistance by actively excreting cytotoxic drugs, thus reducing drug accumulation below the effective level of chemotherapy, leading to MDR. β-elemene can reverse drug resistance by decreasing the function of the ABC transporter substrate, decreasing the ABC protein efflux, and reducing the number of *ABC* genes and protein expression. The mechanism of β-elemene inhibiting the efflux of ABC transporters is shown in Table 2.

Guo et al. reported that β-elemene incubated with other anticancer drugs on ABCB1-overexpressing cancer cell lines significantly enhanced the antitumor effects of colchicine, vinblastine, and paclitaxel. In addition, the study confirmed that apart from a reduction in drug efflux, an increase in drug accumulation, and cytotoxicity, the mechanism of β-elemene in reversing drug resistance involved the inhibition of the efflux of the ABCB1 transporter and the blocking of the substrate efflux function of P-gp [27]. β-elemene reverses the multidrug resistance of human gastric adenocarcinoma cells by increasing the intracellular accumulation of Adriamycin (DOX) and Rhodamine 123(Rho 123) in K562/DNR and SGC7901/ADR cells, and inhibiting the expression of P-gp [27]. Some studies also confirm that β-elemene can reverse drug resistance by inhibiting the substrate efflux function of P-gp [5]. In addition, poly-ADP-ribose polymerase (PARP) is known to be a substrate for the activation of Caspase, acting in the nucleus to repair damaged DNA. Zhang et al. [41] and other studies showed that β-elemene partially reversed the daunorubicin resistance of human myeloid leukemia DNR-resistant strain cells (K562/DNR) by cleaving PARP and downregulating P-gp protein expression, thereby increasing drug accumulation in cells. Li et al. [85] also showed that β-elemene reversed the drug resistance of breast cancer cells by decreasing the expression of P-gp and BCRP resistance proteins.

Natural compounds that reverse MDR by inhibiting P-gp are potential chemotherapeutic MDR modulators. Drug delivery systems have been developed to transport chemotherapeutics to the tumor cell, such as liposomes, nanoemulsions, nanogels, polymer drug conjugates, powders, dendrimers, micelles, and hydrogels [86,87]. Nanoparticles evade and inhibit P-gp efflux activity by being decorated with anti-P-gp materials, such as conjugation or co-encapsulation with traditional anticancer drugs [84]. For example, for overcoming multidrug resistance in leukemia, the antitumor drug mitoxantrone (MTO) and a P-gp inhibitor of β-elemene (βE) were encapsulated at a weight ratio of 1:2 (*w*/*w*) to form MTO/βE-SLNs and were effectively internalized by both K562/DOX and K562 cells through pit-mediated endocytosis. Compared with free MTO, MTO/βE-SLNs induce higher cytotoxicity as they are readily taken up by cells to inhibit intracellular ATP production and P-gp efflux by βE. The decrease in intracellular ATP production leads to the inactivation of the P-gp efflux transporter, an increase in the drug absorption by cells, the prolonged cycle time of MTO and βE, increased plasma half-life, and a significant reversal of drug resistance in leukemia [53]. Zeng et al. developed a nano-drug for the treatment of paclitaxel-resistant lung adenocarcinoma by successfully preparing cabazitaxel liposome, β-elemene liposome, and cabazitaxel-β-elemene complex liposome with good flexibility. Pharmacodynamics studies showed that cabazitaxel liposome and β-elemene liposome were relatively good at overcoming paclitaxel resistance on paclitaxel-resistant lung adenocarcinoma [88]. Compared with cabazitaxel alone, a combination of cabazitaxel and β-elemene has a stronger effect on paclitaxel-resistant lung adenocarcinoma cells, and the effect increases with ratios of β-elemene to cabazitaxel to some extent [88].

In a study investigating the treatment of breast cancer, the MCF-7/DOX fluc cell line with stable overexpression of luciferase was established with potassium d-fluorescein (substrate of ABC transporter). The drug efflux kinetics in MCF-7/DOX fluc cells were monitored, and β-elemene was found to inhibit the function of the ABC transporter by downregulating P-gp, MRP, and BCRP gene and protein expression. DOX and β-elemene enhanced the inhibitory effect of DOX on MCF-7/DOX proliferation, reduced the half-maximal inhibitory concentration (IC50), and had a synergistic anti-proliferation effect [54]. In another study, a combination of elemene and gefitinib (GEF) produced better results compared to single drug treatment in cells with multiple 150-fold resistance to gefitinib. The mechanism showed that elemene enhanced the fluorescence response intensity of D-fluorescein potassium (substrate of ABC drug-resistant protein) in PC-9/GR fluc cells, increases the uptake of GEF by cells, and inhibits the efflux function of drug-resistant proteins [89].

### 3.2. Reduction in Exosomal Transmission

Exosomes are widely studied because they provide a means of intercellular communication. Exosomes may contain microRNAs (miRNAs) that alter chemosensitivity, and the MDR of tumor cells is partly attributed to the intercellular transfer of specific miRNAs. Studies have shown that miRNAs circulate in body fluids in a highly stable and acellular form, which may be due to their incorporation into exosomes [55]. The ability of MDR Breast Cancer Anti-Estrogen (BCA) cells to transmit drug resistance depends on the exosomes released. Studies have confirmed that β-elemene can change the expression of some MDR-related miRNAs, including PTEN and P-gp in MCF-7/ADR and MCF-7/docetaxel (DOC) cells [90]. In addition, it can affect the content of exosomes and reverse drug resistance by reducing exosome transmission [55,90].

Metastasis and drug resistance in cancer are closely related and regulated by multiple factors [91,92]. Studies have shown that exosomes play an important role in EMT [93]. Studies on the role of exosomes in transmission reveal that epithelial–mesenchymal transformed tumor cells can induce EMT of adjacent cells, triggering cell invasion and metastasis. Some studies have found that the role of exosomes in the transmission of drug resistance-related molecules to adjacent cells triggers drug resistance in tumors. Cbl-b can reverse the drug resistance of multidrug-resistant cancer cells and is an important inhibitor of tumor invasion and metastasis in multidrug-resistant gastric and breast cancer cells [94]. Studies have confirmed that P-gp protein delivered by exosomes promotes the drug resistance of sensitive cells in gastric cancer, and the invasion and metastasis of sensitive cells SGC7901/Adr, a cell line with high expression of P-gp, are also closely related to exosome function. In addition, exosomes in gastric cancer downregulate Cbl-b and E-cadherin and upregulate Zincfinger Ebox Binding Homeobox 1 (ZEB-1), ZEB-2, and vimentin in a concentration-dependent manner. miR-1323 is highly expressed in exosomes of multidrug-resistant gastric cancer cells, and miR-1323 inhibits the expression of Cbl-b, leading to the activation of the ERK/Akt-ZEB1/ZEB2 axis, thereby promoting the EMT of gastric cancer sensitive cells and enhancing their invasion and migration ability. Following the incubation of tumor cells with exosomes from SGC7901/Adr cells, a significant increase in drug resistance and metastasis ability was observed, which was affected by the number of exosomes. β-elemene treatment reduced drug resistance and the invasion and metastasis ability of SGC7901 cells treated with SGC7901/Adr exosomes [95]. Therefore, the mechanism of action by β-elemene in the inhibition of invasion and metastasis in gastric cancer-resistant SGC7901/Adr cells is by reversing drug resistance and metastasis caused by exosomes.

### 3.3. Regulation of the Expression of Related miRNA

#### 3.3.1. Through the miR-1323/Cbl-b/EGFR Pathway

A recent study found that β-elemene inhibits the metastasis of multidrug-resistant gastric carcinoma cells through the miR-1323/Cbl-b/EGFR pathway [57]. In a previous study, β-elemene treatment increased Cbl-b expression [72]. Matrix metalloproteases (MMPs) regulate tumor invasion and metastasis by remodeling the extracellular matrix [96]. Mechanistically, β-elemene is reported to inhibit the metastasis of multidrug-resistant gastric cancer cells by regulating the expression of matrix metallopeptidase (MMP)-2/9 and reverse epithelial–mesenchymal transition (EMT). β-Elemene upregulates Cbl-b by inhibiting miR-1323 expression, thereby inhibiting the EGFR-ERK/Akt pathway that regulates MMP-2/9. β-Elemene treatment reversed EMT phenotypes by inhibiting the expression of ZEB1 and ZEB2 in SGC7901/ADR cells. In addition, β-elemene inhibited the ERK/AKT pathway by blocking the ubiquitination and degradation of EGFR, as well as AKT and ERK phosphorylation in a dose-dependent manner. β-Elemene upregulated Cbl-b expression by inhibiting miR-1323, confirming that Cbl-b is a miR-1323 target in MDR gastric cancer cells. miR-1323 targets Cbl-b to prevent Cbl-b-mediated ubiquitination and degradation of EGFR [57]. For further verification, Deng et al. constructed murine metastasis models by tail vein injection of SGC7901/ADR cells, and three groups of a control, a β-elemene group, and a taxol group were treated for in vivo metastasis analysis. Compared with the control group and Apatinib group, the numbers of metastatic tumor nodules were significantly decreased, the expression of miR-1323 was decreased, and Cbl-b and E-cadherin were significantly upregulated in the lungs of nude mice treated with β-elemene, whereas vimentin, MMP-2, and MMP-9 were significantly downregulated. Therefore, β-elemene can reverse EMT in MDR gastric cancer cells by inhibiting the transcription factors ZEB1 and ZEB2 and through the miR-1323/Cbl-b/EGFR pathway [57]. This mechanism is shown in Figure 5.

#### 3.3.2. Increasing PTEN Gene Expression Inhibits the PI3K–AKT Signaling Pathway

An important signaling pathway in the study of multidrug resistance in tumors is the PI3K–AKT signaling pathway [97]. *PTEN* is a common tumor suppressor gene with an inhibitory effect on the PI3K–AKT signaling pathway. Zhang et al. [37] found that β-elemene significantly downregulated intracellular miRNA-29a and miRNA222 in human BCA adriacin (Adr)-resistant MCF-7 cells (MCF-7/Adr) and docetaxel (Doc)-resistant MCF-7 cells (MCF-7/Doc). *PTEN* is the target gene of the two miRNAs, and it is highly expressed following the downregulation of miRNA-29a and miRNA222, thus inhibiting the PI3K–AKT signaling pathway and reversing drug resistance in tumors cells (Figure 5).

### 3.4. Through the Promotion of Apoptosis

Apoptosis is also referred to as programmed cell death, and it eliminates defective cells and maintains a balance between cell proliferation and differentiation [98]. There are two major pathways of apoptosis: the external pathway (mediated by death receptors) and the internal pathway (mediated by mitochondrial membrane potential loss and cytochrome c translocation) [99]. Dysfunction in apoptosis can lead to tumor growth and distant metastasis and is also associated with drug resistance. The pathways by which elemene reverses drug resistance through apoptotic pathways are shown in Figure 6.

#### 3.4.1. Through the Mitochondrial Apoptosis Pathway

The mitochondrial respiratory chain is the main production site of reactive oxygen species (ROS) in the cell. Excessive ROS production may lead to mitochondrial damage and apoptosis [100]. Yao et al. [58] investigated the mechanism of β-elemene resistance to reverse cisplatin (DDP) resistance in lung adenocarcinoma A549/DDP cells. On the one hand, mechanistically, β-elemene promoted A549/DDP cell apoptosis by decreasing mitochondrial membrane potential and increasing intracellular reactive oxygen species (ROS) concentrations, while decreasing cytoplasmic glutathione (GSH) levels, which acted as a scavenger when excessive ROS were produced in the cells [101]. In addition, Bcl-2 is a drug-resistant protein, and a high expression of Bcl-2 has been associated with multidrug resistance [102]. On the other hand, the expression levels of cytochrome c, Caspase-3, and Bad protein (pro-apoptotic protein) increase, while the expression levels of Procaspase-3 protein and Bcl-2 decrease in lung cancer A549/DDP cells. The sensitivity of A549/DDP cells to cisplatin is somewhat improved. Mechanistically, β-elemene increases ROS generation, which decreases mitochondrial membrane potential, imbalances anti-apoptotic and pro-apoptotic members of the Bcl-2 family, and promotes the release of cytochrome c from mitochondria into the cytoplasm, thereby triggering the Caspase cascade that accompanies the cleavage of Procaspase-3 to active Caspase-3 [99]. This series of changes indicate that the mechanism of action of β-elemene as a reversal agent of resistance in A549/DDP cells is through the reduction in mitochondrial membrane potential induced by mitochondrial apoptosis, the activation of an intracellular redox system, and the release of cytochrome c and apoptosis-related genes into the cytoplasm to moderately induce tumor cell apoptosis.

Similarly, β-elemene enhances the cisplatin-induced mitochondrial-dependent apoptotic pathway through the ROS-5’AMP-activated protein kinase (AMPK) signaling pathway and inhibits the proliferation of bladder cancer cells in vitro [59]. A combination of cisplatin and β-elemene increases the cytotoxicity of cisplatin to T24 cells, resulting in the loss of mitochondrial membrane potential, the release of cytochrome c into the cytoplasm, enhanced ROS accumulation, the activation of AMPK, and apoptosis.

β-Elemene has a certain effect on the cell cycle of T24 and 5637 cells, where it decreases the percentage of cells in the S phase and induces G1 phase arrest in a dose-dependent manner. The expression levels of p-STAT3 and Cyclin D1, CDK4, and CDK6 were downregulated, while p21 and p27 were upregulated, and the phosphorylation of Akt (Thr308) decreased. A combination of 20 M cisplatin and 50 g/mL β-elemene was reported to have a greater effect on inducing apoptosis in T24 cells compared to a single drug, decreased mitochondrial membrane potential (ΔΨM), increased the cleavage of Caspase-3 and PARP, increased the expression of Caspase-9, Bax, and cytochrome c, and decreased the expression of Bcl-2. In another study, elemene was reported to reverse the cisplatin-resistant human lung adenocarcinoma cell line (A549/DDP) by regulating the mitochondrial apoptotic pathway [58,103].

#### 3.4.2. Through the Upregulation of Caspase-3 Protein Expression Levels

To investigate the in vivo reversal of the drug resistance efficacy of β-elemene, Yang et al. constructed a cell-derived xenograft (CDX) tumor model with gastric cancer resistant cell line SGC-7901/Adr in nude mice. In vitro, MTT results showed that β-elemene could significantly inhibit the growth of SGC7901/ADR cells. In vivo, compared to the control, tumor volume after elemene treatment was significantly reduced. Immunohistochemical detection of protein expression in tumor tissues showed that the protein expression level of Caspase-3 was significantly upregulated in the tumor tissues of nude mice after β-elemene treatment compared with the control. Therefore, the mechanism of action of elemene to reverse gastric cancer resistance may be to upregulate Caspase-3 protein expression [21].

#### 3.4.3. Decreased Expression of Resistance Genes MDR1, LRP, and TS

Drug resistance is likely to develop during the treatment of bone tumors, and a combination of β-elemene (100 mg/mL) and paclitaxel (20 mg/mL) in the treatment of bone tumors significantly inhibits the growth of U-2OS cells. The mechanism of action of combination therapy includes arresting the cell cycle and inducing apoptosis. U-2OS cells were arrested in G1 and S phases, and the expression levels of *CDK1*, *Cyclin-B1,* and *P27* genes decreased, the expression of *Bcl-2* and *Bcl-w* anti-apoptotic genes decreased, and the expression of pro-apoptotic *Bad* and *Caspase-3* genes was significantly upregulated. The mechanism of action of combination therapy is also related to the inhibition of resistance genes. β-Elemene decreases the expression levels of *MDR1*, *LRP,* and *TS* resistance genes in U-2OS cells.

A combination of β-elemene and paclitaxel was found to inhibit the migration and invasion of tumor cells. In the study of the treatment of β-elemene–paclitaxel on the apoptosis of U-2OS cells, a combination of elemene and paclitaxel was compared with single agents, respectively, and it was found that the combination treatment promoted apoptosis, and gene analysis showed that the combination treatment decreased the expression of Bcl-2 and Bcl-w anti-apoptotic genes, increased the expression of Bad and Caspase-3 pro-apoptotic genes, and decreased the expression of MDR, LRP, and TS genes. In the study of effects of β-elemene on drug-resistant genes for paclitaxel treatment of β-elemene–paclitaxel, the expression of tumor metastasis gene metastasis-associated protein 3 (*MTA3)* and vascular endothelial growth factor (*VEGF)* genes and MTA3 and VEGF proteins was inhibited in U-2OS cells. In the study of the treatment of β-elemene–paclitaxel on tumor angiogenesis-related gene expression, the expression of G-protein coupled receptor 124 (*GPR124*), *MMP-3,* and *MMP-9* genes and protein decreased, while the expression of endostatin, TIMP metallopeptidase inhibitor (*TIMP*)*-1,* and *TIMP-2* increased in U-2OS cells. In tumor-bearing mice, combination therapy is reported to significantly inhibit tumor growth and improve the survival rate of tumor-bearing mice. The mechanism of action is that the number of apoptotic bodies and the expression levels of GPR124, MMP-3, and VEGF increase in tumor tissues. Therefore, β-elemene enhances paclitaxel-induced apoptosis of bone tumor cells by decreasing the expression of resistance genes *MDR1*, *LRP,* and *TS* [60].

#### 3.4.4. Suppressing the STAT3 Signaling Pathway

Gingival squamous cell carcinoma (GSCC) is a rare tumor [104], and cisplatin is a commonly used therapeutic drug, which is highly toxic and prone to drug resistance [105]. The Janus kinase (JAK)/STAT protein family is known to play an important role in the proliferation, apoptosis, invasion, and survival of cancer cells [106]. The JAK/STAT pathway transmits extracellular signals to the nucleus, leading to gene transcription [107]. Among them, STAT3 is tyrosine phosphorylated, transcribed after activation, and with constitutive activity in various cancers [106,108]. Janus-activated kinases (JAKs), including JAK1, JAK2, and JAK3, mediate the phosphorylation of STAT3 [109]. Downstream signaling of STAT3 includes various proteins, such as C-MYC, Mantle Cell Lymphoma (MCL-1), Cyclin D1, Survivin, Caspase, and the Bcl-2 family, regulating cell cycle, apoptosis, and survival [110,111]. The apoptotic process is mediated by Caspase, mainly Caspase-3 and Caspase-9, and the apoptotic pathway can be triggered by the cleavage of Caspase-3. Bcl-2 and Bax play important roles in Caspase-dependent apoptosis. Bcl-2 functions as an anti-apoptotic protein, while Bax promotes apoptosis. Therefore, inhibiting STAT3 activation has potential in cancer therapy, and STAT3 activation is associated with the induction of apoptotic resistance, possibly by increasing Bcl-2 expression [61].

To investigate the anti-tumor effect of β-elemene on GSCC, YD-38 cells were treated with β-elemene at a specific concentration gradient. The results showed that β-elemene inhibited the viability of YD-38 cells in a dose-dependent manner. Cisplatin combined with β-elemene significantly enhanced the inhibitory effect of cisplatin. In addition, the clonogenic assay showed that the combination of a low dose β-elemene group and cisplatin inhibited colony formation. Annexin V-FITC apoptosis assay showed that β-elemene enhanced cisplatin-induced apoptosis in GSCC cells. Western blotting was used to detect the levels of key proteins in the STAT3 pathway in cells treated with cisplatin-combined β-elemene. The results showed a decrease in the expression of p-STAT3, p-JAK2, and Bcl-2, while the expression of Bax and Caspase-3 was significantly increased. This indicated that β-elemene enhanced proliferation, inhibition, and apoptosis by inhibiting the STAT3 signaling pathway. In an in vivo xenograft model, the tumor volume and weight were significantly reduced, and the anti-tumor effect was significantly enhanced in the combination group. Further, compared with cisplatin treatment alone, cisplatin combined with β-elemene decreased the expressions of p-STAT3, p-JAK2, and Bcl-2, and increased the expressions of Bax and Caspase-3 significantly, as well as in the xenograft model [112].

Therefore, β-elemene promotes the anti-proliferative and apoptotic effects of cisplatin in GSCC cells by inhibiting JAK2–STAT3 signaling, inactivating STAT3, regulating downstream apoptotic proteins, downregulating Bcl-2 expression, and upregulating Bax and Caspase-3 [61,112].

#### 3.4.5. Inhibition of CDK8 Gene Expression

Zeng et al. studied the effect of elemene on reversing chemoresistance in lung cancer and its effect on *CDK8* gene expression. C57BL/6 mice were used to detect the CDK8 gene expression in the blood of mice in each group, and a Lewis lung carcinoma mouse model was used to observe the tumor growth and detect the Cdk8 gene expression in the tumor tissues, which were all divided into four groups including a model group, elemene group, doxorubicin group, and combination group for treatment. The results showed that the pretreatment of mice with elemene, DOX, and their combination revealed that the expression level of *CDK8* gene in the serum of mice pretreated with DOX was higher than that of the model group, and the expression level of the *CDK8* gene in tumor tissues of mice in the combination group was lower than that in the DOX group but higher than in the elemene group. The rate of tumor inhibition by combination therapy was higher than in both groups alone, and it was 46.01% in the DOX group and 69.96% in the combination group in the Lewis lung cancer mouse model. In addition, the proliferation rate of A549 lung adenocarcinoma cells cultured in the serum of mice pretreated with DOX and elemene was less than that of the two groups alone in vitro. Therefore, we speculate that the reason for the development of resistance to doxorubicin is the increased CDK8 expression in the treated mice, and elemene can reverse the resistance by inhibiting CDK8. These results suggest that the mechanism by which elemene reverses chemoresistance in lung cancer may be related to its inhibition of *CDK8* gene expression [75].

#### 3.4.6. Inhibition of P21 Gene Expression

Elemene has been found to reverse sorafenib resistance and synergize with sorafenib. Considering the mechanism of action, elemene can reverse hepatoma HepG2 cells, causing changes in Raf kinase and P21 protein levels [76,113]. Zhang et al. studied the effect of elemene on reversing chemoresistance in lung cancer and its effect on *P21* gene expression. P21 is a member of the CDK inhibitor (CKI) family of cyclin-dependent kinase inhibitors, while CDK8 is a member of the cyclin-dependent kinase CDK [113]. Study showed that the pretreatment of mice with elemene, DOX, and their combination showed no significant difference in serum *P21* gene expression level between the elemene combined with DOX pretreatment group, the two pretreatment groups alone, and the normal group in vivo. However, the expression level of the *P21* gene in tumor tissues of mice in the combined group was lower than in the DOX group but higher than in the elemene group. After treatment, the tumor inhibition rate of DOX and elemene combined with DOX was significantly higher than the two groups alone. In vitro, the proliferation rate of A549 lung adenocarcinoma cells cultured in the serum of mice in the combined pretreatment group was lower than that of the two groups alone. Therefore, elemene was found to play a role in reversing chemotherapy resistance in lung cancer, and its mechanism may be related to the inhibition of *P21* gene expression. Similarly, in vitro, some studies have shown that elemene can reverse sorafenib resistance and synergize with sorafenib to exert a combined antitumor effect, the mechanism of which may be related to the reversal of Raf kinase and P21 protein levels in hepatoma HepG2 cells [76,113]. Elemene not only reverses the chemoresistance of metastatic lung cancer by inhibiting the expression of the *P21* gene and protein [114], but also inhibits the expression of CDK8, and the growth of tumors. This suggests that elemene can prevent chemoresistance in lung cancer by inhibiting CDK8-regulated P21-mediated paracrine activities [115].

#### 3.4.7. Destruction of DNA Repair Activity and Activation of Apoptotic Signaling Pathways

β-Elemene is reported to enhance the sensitivity of resistant ovarian cancer cells to cisplatin by downregulating excision repair cross-complementation group-1 (ERCC-1) and X-linked inhibitor of apoptotic protein (XIAP) and inactivating c-Jun NH2-terminal kinase (JNK). In A2780/CP70 cells, the dose-regulating factors for cisplatin range from 35 to 60, and 1.6 to 2.5 for A2780 cells. In cisplatin-resistant ovarian cancer cells, β-elemene inhibits cisplatin-induced expression of *ERCC-1*, a marker gene for the repair of cisplatin-induced DNA damage in the nucleotide excision repair pathway. In addition, β-elemene not only decreases the level of XIAP but also downregulates cisplatin-mediated XIAP expression in chemo-resistant cells. Furthermore, β-elemene blocks a cisplatin-stimulated increase in phosphorylated JNK levels in these cells. These findings suggest that β-elemene enhances cisplatin sensitivity in human drug-resistant ovarian cancer cells in part by disrupting DNA repair activity and activating the apoptotic signaling pathways [62].

β-Elemene reduces the DNA repair ability of tumor cells. Platinum compounds bind to DNA and cause DNA damage, leading to tumor cell apoptosis as the main anti-tumor mechanism. Elevated nucleotide excision repair (NER) is an important mechanism by which cells develop platinum compound resistance. High levels of the ERCC-1 protein support efficient NER. Li et al. [116] found that β-elemene decreased the expression of the *ERCC-1* gene and protein in ovarian cancer cells, thus making cells more sensitive to platinum compounds [117] and inducing tumor cell apoptosis.

#### 3.4.8. Inhibition of WEE1 Expression

Apoptosis is regulated by the division cycle, which is dominated by the complex cyclin-dependent kinases (CDKs)/Cyclins formed by the binding of CDKs and cyclins. The activation of the CDKs/Cyclins complex causes cells to move from G2 to M phase, and WEE1 kinase regulates this process [118]. WEE1 protein kinase is a key enzyme regulating cells from G2 to M phase. Abnormal expression of WEE1 can cause cell division arrest in the G2 phase, induce DNA damage, and lead to apoptosis or carcinogenesis [119]. The inhibition of WEE1 protein expression can reverse the resistance of tumor cells to chemotherapeutic drugs. Wu et al. studied the effect of β-elemene on VCR resistance of colon cancer cells and found that vincristine (VCR) inhibited the proliferation of SW-480 cells in a dose-dependent manner. Therefore, β-elemene could enhance the inhibitory effect of VCR on SW-480 cells, inhibit WEE1 protein expression in a dose and time-dependent manner, enhance the ability of VCR to chemosensitivity of colon cancer cells, and play a pro-apoptotic role. The combined intervention of β-elemene + VCR inhibits cell proliferation activity. In another study, the pcDNAWEE1 plasmid was constructed to upregulate WEE1 protein expression in cells. It was found that the overexpression of WEE1 inhibited the combined effect of β-elemene + VCR. β-Elemene was found to induce the sensitivity of colon cancer cells to VCR by inhibiting WEE1 expression [77].

The mechanism of elemene reversing multidrug resistance by inducing the apoptosis of tumor cells through various pathways is shown in the Figure 6. Elemene inhibits ABCB1 transporters (BCRP and P-gp) on cell membranes, reduces drug efflux, decreases exosome delivery, decreases the expression of resistance genes *MDR1*, *LRP,* and *TS*, upregulates Caspase-3 protein expression, induces apoptosis through the mitochondrial apoptotic pathway, inhibits the STAT3 signaling pathway, and inhibits *CDK8* and *P21* gene expression.

### 3.5. Regulation of the Cell Cycle Pathway

The eukaryotic cell cycle is regulated by a series of CDKs, whose activity is positively regulated by cyclins and negatively regulated by cyclin-dependent kinase inhibitors (CKIs). In drug-resistant ovarian cancer cells, β-elemene enhances the growth inhibition of cisplatin-induced drug resistance. However, the proportion of cells in the G2–M phase increases significantly, leading to G2–M cell cycle arrest. In addition, the G2–M arrest of the cell cycle is associated with the changes in cell cycle regulatory molecules in drug-resistant ovarian cancer cells. By detecting the expression of intracellular proteins, the expression of Cyclin B1 and Cdc2, which regulate the cell cycle at the G2–M boundary, was downregulated. Further experiments showed that β-elemene and cisplatin increased the expression of p53, p21waf1/cip1, and growth arrest and DNA damage-inducible protein45 (Gadd45) proteins in human ovarian cancer cells. The overexpression of these proteins may be an important reason for the decrease in Cdc2–cyclin B1 activity and G2 checkpoint control in ovarian cancer cells.

Tyrosine phosphatase Cdc25C activates Cdc2 by the dephosphorylation of tyrosine 15 (Tyr15) residues phosphorylated by Wee1 kinase. Studies report that the causes of Cdc2 inactivation include an increase in Wee1 kinase phosphorylation and a decrease in Cdc25C dephosphorylation. Increased Cdc2 and Cdc25C phosphorylation decreases Cdc2–cyclin B1 activity. These results suggest that β-elemene increases the sensitivity of resistant ovarian cancer cells to cisplatin by regulating the cell cycle G2 checkpoint and inducing cell cycle G2–M arrest [63]. The mechanism of elemene reversing the multidrug resistance of tumor cells by blocking the cell cycle is shown in Figure 7.

### 3.6. Induction of Autophagy

In exploring the effect of elemene on the DDP resistance of lung adenocarcinoma cells, elemene was found to reverse the drug resistance of SPC-A-1/DDP cells by promoting Beclin-1 (a key regulator of autophagy)-induced autophagy. The overexpression of Beclin-1 and elemene treatment have similar effects on autophagy and autophagic apoptosis of SPC-A-1/DDP cells. Elemene improves the apoptosis and drug sensitivity of SPC-A-1/DDP, a lung cancer cell line resistant to human DDP, by inducing Beclin-1 expression. The sensitivity of human lung adenocarcinoma cell line SPC-A-1 and its anti-DDP strain SPC-A-1/DDP to elemene is similar. Low doses of elemene increase the sensitivity of SPC-A-1/DDP cells to DDP, the expression of multidrug resistance proteins and cell proliferation decrease, and cell autophagy and autophagic apoptosis increase. In contrast, the Beclin-1 combination can reduce elemene-induced apoptosis of autophagic cells and counteract elemene-induced sensitivity of SPC-A-1/DDP cells [64].

In another study, β-elemene enhances 5-Fu sensitivity in p53 wild-type colorectal cancer cells by significantly inhibiting cell proliferation in a concentration-dependent manner, reversing 5-Fu resistance by inducing death-promoting autophagy and Cyclin D3-dependent cycle arrest. In vivo studies demonstrated that 5-FU and β-elemene combination treatment can significantly inhibit the tumor volume, and the effect is higher than that of the single agent in the HCT116p53^−/−^ all line xenograft model constructed from 5-fluorouracil-resistant p53 deficient colorectal cancer HCT116p53^−/−^ cells [65] (Figure 5).

### 3.7. Inhibition of Stemness

#### 3.7.1. Downregulation of Enhancer of Zeste Homolog 2 (EZH2)

Studies have found that the existence of some self-renewing cancer stem cell (CSC) populations in tumors is closely related to the development and metastasis of tumors. In cancer therapy, cancer stem cell populations are resistant to chemotherapy and this resistance is intrinsic. This suggests that surviving cancer stem cells can reproduce after chemotherapy and subsequently cause tumors [120]. These cancer stem cell populations are protected in vivo, and the main mechanism of stem cell protection is through the expression of ABC transporters. These transporters are guardians of stem cell populations in vivo and they protect them from adverse effects of chemotherapy [121].

Some inhibitors are now being used to circumvent the role of the transporters in protecting cancer stem cells. In the treatment of NSCLC, the most commonly used drug is gefitinib, which is used as an EGF receptor tyrosine kinase (EGFR-TKIS) inhibitor to treat NSCLC patients sensitive to EGFR through targeted therapy. However, long-term use leads to drug resistance, resulting in reduced effectiveness. In addition, studies have shown that failure to kill cancer cells by chemotherapy inevitably leads to the enrichment of cancer stem cells (CSCs) [122]. CSCs are small tumor cell subpopulations that may be derived from neoplasms, but adopt stem-like characteristics, including enhanced motility and aggressiveness, the transition from epithelial to mesenchymal phenotypes, resistance to tumor drugs, and tumor initiation potential [123]. Therefore, the induction of stem cell-like cells is considered a risk factor for poor prognosis and cancer recurrence, and targeting CSCs presents a promising new therapeutic strategy [124].

Studies have demonstrated that the combination of β-elemene and chemotherapeutic drugs has stronger antitumor activity and reverses the resistance to EGFR-TKI in cancer cells A549 and H1299. The combined treatment inhibits the G2/M phase cells, and significantly reduces the migration ability of A549 and H1299 cells. Further experiments show that combination therapy increases the level of epithelial marker E-cadherin and inhibits the expression of mesenchymal markers ZEB1, Snail, and Twist, indicating that the mechanism of action is related to EMT [125]. The activation of the EMT program allows cancer cells to acquire stem-like characteristics. Combination therapy downregulates stem cell-related genes including sonic hedgehog (SHH), hairy, and enhancer of split (HES)1, NOTCH1, C-MYC, Ry-Box (SOX)2, and octamer-binding transcription factor (OCT4), and decreases the percentage of CD44^+^CD24^−/low^ cell populations with a stem phenotype [66,126].

The synergistic effect of olefin and gefitinib on NSCLC cells depends in part on the downregulation of EZH2, an H3K27 methyltransferase recognized as a gene transcription regulator and cancer driver primarily through its epigenetic and global ability to modify tumor-related genes [78,79]. Notably, elemene and gefitinib combinations were found to cause a significant decrease in EZH2 levels in both A549 and H1299 cells, compared with single drugs. The use of lung cancer cells in conjunction with the established EZH2 inhibitor GSK343, a key factor in the stem-cell targeting of combination therapies, resulted in a significant reduction in cell motility and migration. Furthermore, to evaluate elemene drug efficacy in vivo, the A549 subcutaneous xenografts was established. The in vivo results showed that the tumor volume decreased by 48% in the gefitinib group and 59% in the elemene group compared with the control group, whereas the combined treatment decreased the tumor volume by 82%. In addition, the levels of EZH2 and PD-L1 were repressed, and the relative levels of CXCL9 and CXCL10, which may be subjected to EZH2-dependent epigenetic modification, were remarkably upregulated in tumors with combined treatment. Thus, mechanistically, elemene and gefitinib combination therapy inhibits EZH2 and PD-L1 levels and the cancer-stem phenotype, reverses the mesenchymal phenotype, and inhibits their ball-forming ability, thereby inhibiting tumor development in xenograft mice [66].

#### 3.7.2. Inhibition of Stemness

In a study on the effect of β-elemene on MCF-7/ADM (Adriamycin-resistant cells) breast cancer stem cells (BCSC), the expression of P-gp and BCRP in MCF-7/ADM cells was 77.78 ± 9.55% and 32.33 ± 5.12%, respectively, and the CD44^+^ CD24^−/low^ cell rate was 64.79 ± 11.78%, which were significantly higher than in MCF-7/S cells (3.97 ± 1.51, 14.26 ± 2.51, 18.79 ± 3.28%, respectively). Therefore, elemene was found to reduce the CD44^+^ CD24^−/low^ cell ratio and reverse breast cancer resistance to Adriamycin by reducing the serum pellet-free rate of MCF-7/ADM cells and the expression of *BCRP* and *P-gp* genes/proteins [67].

Similarly, in glioblastoma stem-like cells (GSCs), β-elemene decreases the formation of GSC spheres, disperses GSC sphere formation, and inhibits GSC proliferation both in vitro and in vivo. An in vivo experiment showed that β-elemene inhibited the development of tumors with β-elemene treatment in nude mice transplanted with GSCs. Immunohistochemical analysis of tumors showed that the expression of stem cell markers CD133 and ABCG2 (also known as BCRP1) is significantly downregulated and the differentiation marker glial fibrillary acidic protein (GFAP) expression increases in the GSC spheres and xenografts treated with β-elemene. Further experiments show that the sensitivity of GSCs to temozolomide (TMZ) is enhanced with β-elemene and TMZ treatment compared with the control or the TMZ groups in vitro and in vivo. These results suggest that β-elemene inhibits the stemness of GSC spheres, promotes differentiation, and sensitizes GSCs to TMZ [68] (Figure 5).

### 3.8. Inhibition of Epithelial–Stromal Transformation

The treatment of metastatic colorectal cancer (mCRC) is limited. Chemotherapy combined with anti-epidermal growth factor receptor (EGFR) or VEGF monoclonal antibodies is mainly used, but this treatment modality has numerous limitations and is only effective for mCRC. In addition, its effectiveness is limited in RAS wild-type patients because of KRAS mutations. A previous study found that the combination of natural drug β-elemene and cetuximab had better efficacy and showed the best synergistic effect. The combined treatment of β-elemene (125 g/mL) and cetuximab (25 g/mL) in KRAS mutant CRC cells HCT116 and Lovo showed a significant reduction in the activity of CRC cells, cell cycle arrest in the G0/G1 phase, and anti-proliferative effects.

Epithelial–mesenchymal transformation (EMT) is known to play an important role in development, metastasis, and drug resistance in cancer [127]. In a previous study, treatment with a combination of β-elemene and cetuximab increased the sensitivity of KRAS mutant CRC cells (HCT116 and Lovo cells) by inducing ferroptosis and inhibiting EMT. Ferroptosis is a new form of regulated cell death caused by the failure of the glutathione-dependent lipid-peroxide-scavenging network [128]. Combination therapy triggered several ferroptosis-related events including reactive oxygen species (ROS) accumulation, glutathione (GSH) depletion, and lipid peroxidation [129]. Elsewhere, treatment with a combination therapy increased the expression of Heme Oxgenase-1(HO-1) and transferrin, but decreased the expression of proteins that negatively regulate ferroptosis (GPX4, SLC7A11, FTH1, glutaminase, and SLC40A1) in HCT116 and Lovo cells [130]. In addition, the combination treatment reduced the expression of mesenchymal markers (vimentin, N-calcium-adhesion protein, Slug, Snail, and MMP-9), but promoted the expression of epithelial marker E-calcium-adhesion protein, indicating that it may inhibit the cell migration of KRAS mutant CRC cells and inhibit EMT. It has been reported that therapy-resistant cancer cells undergoing EMT are more likely to be killed by ferroptosis inducers. An in situ CRC animal model with the implantation of luciferase-transfected HCT116 cells into the mesentery was established and tumor growth was monitored by an IVIS Lumina LT imaging system. In an orthotopic murine colon cancer model, treatment with cetuximab alone increased the tumor volume and lymph node metastases, indicating that KRAS mutant CRC cells exhibit resistance to cetuximab. However, the combined treatment inhibited tumor growth and lymph node metastasis in KRAS mutants. Immunohistochemical results showed that after combined treatment with elemene and cetuximab, a low expression of GPX4, SLC7A11, and SLC40A1 and a high expression of transferrin were found in KRAS mutant CRC samples. Moreover, the expression of E-cadherin increased and Vimentin expression decreased with combined treatment. Therefore, β-elemene can be used as a novel pro-mast disease inducer to inhibit the migration of KRAS-mutated CRC cells and enhance the sensitivity of KRAS-mutated colorectal cancer cells by combining with cetuximab to induce ferroptosis and inhibit EMT [69]. The mechanism of β-elemene inhibition of epithelial–mesenchymal transformation is shown in Figure 8.

### 3.9. Upregulated Expression of Estrogen Receptor α(ERα)

Endocrine therapy is an important treatment for estrogen receptor (ER)-positive breast cancer. However, many of the endocrine therapies fail when the tumor loses its ER expression during treatment. The loss of ER expression is the main cause of Tamoxifen (TAM) resistance in MCF-7 cells. To solve this problem, Zhang et al. [70] created the TAM drug-resistant cell line MCF-7/TAM using an ER-negative MCF-7 breast cancer cell line. The sensitivity of MCF-7/TAM cells to TAM was restored by elemene treatment through the upregulation of ER mRNA levels in the cells, which in turn upregulates ER expression. ER can also be re-expressed by reducing the protein expression levels of Ras, MEK1/2, and P-ERK1/2 in MCF7/TAM cells. Therefore, elemene reverses TAM resistance by upregulating ER mRNA and the re-expression of ER via the MAPK pathway [70] (Figure 5).

## 4. Other Pathways That Reverse Resistance

### 4.1. Inhibition of the c-Met Signaling Pathway

Chen et al. investigated the mechanism of elemene in reversing the resistance of lung cancer PC-9 cells to gefitinib. PC-9 cells induced by Hepatocyte Growth Factor (HGF) were treated with different concentrations of gefitinib and elemene alone or in combination. The levels of c-Met, AKT, and their phosphorylated proteins in PC-9 cells were measured to determine the cell viability, calculate the IC50, and examine the effect of elemene on the invasion ability of HGF-induced gefitinib-resistant PC-9 cells. The results showed that elemene could significantly inhibit the viability of lung cancer PC-9 cells (*p* < 0.05), and the growth inhibition rate of elemene on PC-9 cells increased significantly with an increase in the drug dosage. A combination of c-Met inhibitor SU11274 with gefitinib on HGF-induced PC-9 cells significantly decreased the survival rate compared to that of gefitinib alone on HGF-induced PC-9 cells. Elemene and HGF combined with gefitinib significantly inhibited the invasive ability of lung cancer PC-9 cells and upregulated the protein levels of p-Met and p-AKT. Therefore, we speculate that the mechanism of elemene in reversing the resistance of lung cancer PC-9 cells to gefitinib may be related to the inhibition of c-Met and its downstream signaling pathways activated by HGF [80].

### 4.2. Upregulation of E3 Ubiquitin Ligase, c-Cbl, and Cbl-b

Phosphatidylinositol 3 kinase (PI3K) signaling can control cell growth by activating downstream pathways mediated by serine/threonine-protein kinase AKT. The inhibition of PI3K/Akt signal transduction can reduce the growth of tumor cells and inhibit tumor formation. To investigate the ability of β-elemene to reverse drug resistance to DOX in vivo, an ADR-resistant gastric adenocarcinoma SGC7901/ADR xenograft model, which is a drug resistance model with expression of P-gp, was established in nude mice. The in vivo results showed that neither DOX nor β-elemene significantly inhibited the growth of tumor xenografts; however, the combination of DOX and β-elemene significantly inhibited the growth of SGC7901/ADR xenografts in nude mice, with an inhibition rate of 42.6% in the combination group. In addition, β-elemene (1, 5 and 10 ug/mL) significantly increased in Rho 123 and DOX mean fluorescence intensity (MFI) in DNR-resistant leukemia cell lines K562/DNR and SGC7901/ADR. Treatment with elemene for 24 h resulted in a significant decrease in phosphorylated Akt. After SGC7901/ADR cells were treated with elemene for 24 h, the c-Cbl and Cbl-b proteins in the cells were significantly upregulated. These showed that β-elemene can inhibit the drug transport activity of P-gp. Mechanistically, β-elemene can inhibit PI3K/Akt signaling to reverse P-gp-mediated MDR by upregulating E3 ubiquitin ligase Cbl Proteins (c-Cbl, and Cbl-b) and enhancing the efficacy of DOX, daunorubicin, and epirubicin in K562/DNR and SGC7901/ADR cells [72].

### 4.3. Inhibition of the ERK1/2-Bcl-2/Survivin Pathway

Glial maturation factor (GMF) is a 17 kDa intracellular stress-related signal transduction regulator [131]. GMFβ can inhibit the growth of C6 in rats and block the G0/G1 cell cycle of human HG-1 glioblastoma cells in vitro [131]. The behavioral response of cells to extracellular stimuli, such as growth factors and hormones, is also regulated by the MAPK pathway [57]. ERK1/2 is a member of the MAPK family, and the constitutive activation of the ERK1/2 pathway plays an important role in cell proliferation and drug resistance in glioblastoma and ovarian cancer [132]. To investigate the mechanism of elemene inhibition of the proliferation of human U87 and U251 glioma cells, the expression of GMFβ was downregulated in U87 cells treated with β-elemene under RNA interference. Further, β-elemene inhibited the proliferation of U87 glioblastoma cells through the GMFβ-dependent inactivation of the ERK1/2-Bcl-2/Survivin pathway. The inhibitory effect of ERK1/2 inhibitor PD98059 on ERK1/2 enhanced the anti-tumor effect of β-elemene and reduced the expression levels of Bcl-2 and Survivin. β-Elemene also increased the sensitivity of U87 glioblastoma cells to chemotherapy TMZ, which was synergistically enhanced by PD98059. In conclusion, these results suggest that the antiproliferative effect of β-elemene on glioblastoma is through the ERK1/2-Bcl-2/Survivin pathway, which is dependent on the inactivation of GMFβ. Therefore, β-elemene inhibits proliferation through crosstalk between GMFβ and ERK1/2, weakens the resistance of glioblastoma cells to temozolomide, and functions as a good chemosensitizer against TMZ in glioblastoma brain tumors [73].

### 4.4. Prevention of CTR1 Degradation

In the treatment of hepatocellular carcinoma (HCC), chemotherapeutic platinum drugs are susceptible to drug resistance, which is mainly due to the reduction in platinum entering cells. The role of platinum drugs is related to their binding to DNA and plays a cytotoxic role mainly through DNA damage. CTR1 is the main controller of intracellular platinum accumulation and controls the cytotoxic effect of platinum chemotherapeutic drugs [133]. Studies have shown that β-elemene has little effect on the proliferation of HCC cells but plays a synergistic anti-proliferative role when combined with oxaliplatin. In vivo and in vitro experiments reveal that β-elemene enhanced the sensitivity of HCC cells to oxaliplatin by upregulating CTR1 [74]. In this study, to determine whether β-elemene led to enhanced oxaliplatin uptake, the level of platinum bound to DNA in MHCC97H was measured, and it was found that the combination of β-elemene with oxaliplatin had a synergistic anti-proliferative effect compared with the oxaliplatin alone treatment. Compared to oxaliplatin alone, a combination of β-elemene and oxaliplatin produced about 2.50, 2.10, and 2.08 times more DNA adducts in MHCC97H cells containing 4, 8, and 16 μm oxaliplatin, respectively, which significantly increased the platinum-DNA adduct, thereby increasing platinum accumulation. Protein analysis showed that a combination of oxaliplatin/β-elemene greatly reduced the expression of anti-apoptotic proteins Bcl-2 and Bcl-XL and upregulated the expression of pro-apoptotic protein BAX, which promoted the release of cytochrome c from the mitochondria into the cytoplasm and played a pro-apoptotic role. More importantly, the analysis of CTR1 siRNA showed that β-elemene could increase the intercellular platinum content by stabilizing CTR1, thereby counteracting the degradation of CTR1 after oxaliplatin treatment [74]. Therefore, β-elemene prevents a decrease in CTR1 caused by oxaliplatin treatment and plays a synergistic anti-proliferative role by enhancing the absorption of oxaliplatin by cells.

### 4.5. Inhibition of PI3K/Akt/mTOR Signaling Pathway

Abnormal activation of the PI3K/Akt/mTOR signaling pathway is an important factor in tumorigenesis and development. Phosphatidylinositol 3 kinase (PI3K) signaling can control cell growth by activating downstream pathways mediated by serine/threonine-protein kinase AKT. The inhibition of PI3K/Akt signal transduction can reduce the growth of tumor cells and inhibit tumor formation. The regulatory subunit of PI3K binds to upstream signaling proteins and catalyzes PIP2 phosphorylation to PIP3, which activates PDK1 and phosphorylates AKT, further activating the downstream mTOR signaling pathway [134]. The mTOR signaling pathway is one of the three super-signaling pathways of EGFR (PI3K/Akt/mTOR, ERK, STAT3 pathways). Sustained activation of this pathway is due to mutations in the *EGFR* gene, through which EGFR-TKI can exert its inhibitory effect. The reason for EGFR-TKI resistance is that a component of this pathway is altered in NSCLC patients with mutations in the *EGFR* gene, which leads to sustained activation of the pathway independent of EGFR [116]. Therefore, the phosphorylation level of the PI3K/Akt/mTOR pathway can reflect a change in the cell resistance level to some extent. Studies have shown that elemene injectable emulsion inhibits the expression and function of p-PI3K, p-Akt, and p-mTOR resistance proteins in gefitinib-resistant NCI-H460 cells. Therefore, we speculate that elemene injectable emulsion may affect and inhibit the phosphorylation of the PI3K/Akt/mTOR pathway, thereby reversing the resistance of NCI-H460 to gefitinib [81].

## 5. Discussion

Elemene is a sesquiterpene compound with anticancer activity, extracted from traditional Chinese medicine *Curcuma zedoaria* and autonomously developed as a drug in China. Both elemene alone and elemene in combination with chemotherapy play a role in the treatment of some drug-resistant cancers, such as doxorubicin-resistant breast cancer, cisplatin-resistant lung cancer, Adriamycin-resistant gastric cancer, and so on. Some studies found that the long-term effect of elemene emulsion failed to induce the expression of messenger ribonucleic acid (mRNA) and P-glycoprotein (P-gp) of multidrug resistance gene 1 (MDR1) in NEL-7402 cells [135]. Elemene is not closely related to MDR1 gene expression and is not prone to developing multidrug resistance, perhaps indicating that elemene is somewhat helpful in the treatment of drug-resistant cancers [135].

Studies have shown that elemene can reverse drug resistance at the cellular level in vitro, at the animal level in vivo, as well as at the clinical level. At the cellular level in vitro, elemene exerted a reversal of drug resistance by promoting apoptosis in drug-resistant cells, promoting iron death, reducing exosome delivery, inducing autophagy, regulating related target proteins and miRNAs, regulating the cell cycle, and inhibiting EMT and stemness. β-Elemene was found to inhibit the function of the ABC transporter by downregulating P-gp, MRP, and BCRP gene and protein expression [54], and P-gp efflux transporter inactivation results from reduced intracellular ATP production [53]. β-Elemene can reduce the delivery of exosomes bound by some MDR-related miRNAs (PTEN, P-gp, and miR-1323) [90,95]. Elemene can make PTEN highly expressed by downregulating miRNA-29a and miRNA222, thereby inhibiting the PI3K–Akt signaling pathway [37].

At the animal level in vivo, elemene inhibited tumor growth in the CDX tumor model of gastric cancer resistant SGC-7901/ADR cells and the Lewis lung cancer mouse model [21]. Elemene reduced metastatic tumor nodules in a mouse metastasis model constructed with SGC7901/ADR cells and reduced the expression of miR-1323, Cbl-b, and E-cadherin in the lungs of nude mice, whereas vimentin, MMP-2, and MMP-9 were significantly downregulated [21,57]. β-Elemene upregulated Caspase-3 protein expression in tumor tissues of nude mice in a gastric cancer drug resistance model [21]. Cisplatin combination β-elemene enhanced the gingival squamous cell carcinoma antitumor effect, decreased the expression of p-STAT3, p-JAK2, and Bcl-2, and significantly increased the expression of Bax and Caspase-3 in vivo [112]. Elemene combined with chemotherapy decreased CDK8 gene expression levels in tumor tissues of a Lewis lung cancer mouse model [75]. β-Elemene downregulated stem cell marker BCRP1 expression and increased GFAP expression in the GSC spheres and xenografts [68]. Combination treatment with 5-FU and β-elemene significantly suppressed tumor volume in a xenograft model of HCT116p53-/- cells, a p53 deficient colorectal cancer that was resistant to 5-FU [65]. In an orthotopic mouse colon cancer model, elemene and cetuximab combination treatment inhibited tumor growth and lymph node metastasis in a KRAS mutant, inhibiting EMT [69].

At the clinical level, for patients with chemotherapy-resistant advanced gastric cancer, elemene emulsion combined with chemotherapy can improve the efficacy and increase the sensitivity of tumor cells to chemotherapy drugs, and reverse the resistance to chemotherapy drugs [22].

This review summarizes the effects of elemene on many mechanistic pathways involved in the reversal of drug resistance, demonstrating that elemene functions as a multi-target to exert network pharmacological effects as a modern traditional Chinese medicine. Therefore, we conclude some targets for the possible role of elemene in reversing drug resistance.

β-Elemene reverses multidrug resistance mediated by the ABCB1 transporter. Docking analysis techniques indicated that the interaction of β-elemene with the ABCB1 transporter was through docking or binding to site-1 of ABCB1 [27]. 1p5e is a complex of Cyclin A and Cyclin-dependent protein kinase2 (CDK2) (Cyclin A–CDK2). The aberrant upregulation of Cyclin A–CDK2 activity is an important biological feature of tumor cells, and in some cancer cells, extremely active Cyclin A–CDK2 activity is frequently detected, and Cyclin A–CDK2 has emerged as an important target for the treatment of cancer. Molecular docking indicated that β-elemene interacts with 1p5e via hydrocarbyl and π-alkyl bonds; it is predicted that β-elemene molecules can act on the active aberrant Cyclin A–CDK2 [136]. However, the binding mode by which elemene compounds act on other targets is unknown. To further define the mechanism of action of elemene, how elemene acts on target proteins and target RNAs should be studied in depth. In the apoptosis pathway, ROS in the mitochondria may be a key target of elemene action as β-elemene increases ROS generation, which decreases mitochondrial membrane potential, imbalances anti-apoptotic and pro-apoptotic members of the Bcl-2 family, and promotes the release of cytochrome c from mitochondria into the cytoplasm, thereby triggering the Caspase cascade that accompanies by the cleavage of Procaspase-3 to active Caspase-3 [58]. In another mechanism, the promotion of ROS production by elemene may also activate the AMPK signaling pathway to promote apoptosis [59].

In the treatment of MDR gastric cancer cells, we speculate that miR-1323 may be a functional target of elemene to reverse drug resistance, and elemene inhibited miR-1323, which in turn upregulated Cbl-b expression and then inhibited the EGFR-ERK/Akt pathway. β-Elemene can also mediate and downregulate miRNA-29a and miRNA222, thereby inhibiting the PI3K–AKT signaling pathway [37]. In drug-resistant breast cancer, β-elemene reverses drug resistance by mediating MDR-related miRNA (miR-34a and miR-452), P-gp, and PTEN genes in exosomes [90]. Therefore, the targets of elemene acting on miRNA may include miR-1323, miRNA-29a, miRNA222, miR-34a, and miR-452. In addition, β-elemene can inhibit JAK2–STAT3 signaling, thereby regulating cell survival and apoptotic downstream proteins of STAT3, including Bcl-2, Bax, and Caspase-3 in GSCC. Therefore, STAT3 might be a critical target of β-elemene. β-Elemene acts on GMFβ to inactivate GMFβ, which in turn inactivates the ERK1/2-Bcl-2/Survivin pathway, which is dependent on GMFβ activation. GMFβ may be a putative target of β-elemene [73]. β-Elemene can stabilize the transporter CTR1 of platinum drugs into cells, increasing the amount of intercellular platinum, thus counteracting the degradation of CTR1 after oxaliplatin treatment [74].

In addition, elemene can contribute to improving the immune function of patients, including cellular immunity and humoral immunity. Treatment by intraperitoneal perfusion with elemene in colorectal cancer patients showed that compared with the control group, the CD4^+^/CD8^+^ T cells as well as NK cells gradually increased in the elemene-treated group, suggesting that elemene contributes to the improvement of cellular immune function, and that the IL-2 level can be obviously increased by the elemene-treated group, while exerting an inhibitory effect on IL-10 and TNF-α, indicating that elemene elevates the humoral immune function [137]. IL-2 can play immunoregulatory as well as antitumor roles by inducing the expression of immune cells such as B cells [138]. As an inhibitory cytokine, IL-10 would suppress antitumor immunity [139]. As an inflammatory factor, TNF-α is a major immunomodulatory and proinflammatory factor and TNF-α would enhance the adhesive capacity of peritoneal mesothelial cells and promote tumor cell adhesion [140]. In addition, studies have found that P-gp on lymphocytes is caused by the activation of cytokines such as TNF-α [141], and TNF-α can upregulate the expression of P-gp [142], which indicates that the reversal of drug resistance may be related to the regulation of immunity. Elemene may inhibit the expression of P-gp by inhibiting the level of TNF-α, thereby reversing drug resistance. Similarly, in the treatment of NSCLC patients, the levels of CD3^+^, CD4^+^, and CD8^+^ cells were higher in the elemene injection plus docetaxel and nedaplatin treatment groups compared with the control group treated with docetaxel plus nedaplatin [143]. In addition, in a mouse H22 liver cancer model, after intraperitoneal injection of β-elemene, the T-lymphocyte turnover index was examined, and it was found that elemene can increase the T-lymphocyte turnover rate in tumor-bearing mice in vitro, and can promote the generation of T-lymphocytes, thereby improving the body’s immune capacity [144]. Therefore, elemene can both reverse drug resistance and affect the immune system, but the reversal of drug resistance by immune pathways has been less reported, and it will be possible in the future to explore whether elemene can reverse drug resistance through pathways that regulate immunity.

According to the mechanisms of elemene in reversing drug resistance as reviewed in this paper, we believe that elemene plays a multitarget role in reversing drug resistance and has an important role in the treatment of MDR. Therefore, we speculate that elemene may become a potential drug for reversing drug-resistant cancers in the clinic. At present, there are many reports on the reversal of drug resistance by elemene, and the mechanism of resistance reversal by elemene is gradually clear with the depth of research, but there are fewer studies on the binding mode of elemene compounds acting on targets. In order to gain more insight into the mechanism of elemene reversal of drug resistance, the mechanisms of the reversal of drug resistance associated with immunity and the binding of elemene to targets should be more investigated. The drug resistance of elemene should be further investigated in the clinic.

## 6. Conclusions

In conclusion, multidrug resistance in chemotherapy has been widely studied, and especially natural drugs that can treat tumors and reverse multidrug resistance have become a hotspot in anti-tumor research. Elemene, a natural drug, is a new anticancer drug with independent intellectual property rights in China. It has a broad-spectrum antitumor effect and is less toxic. In vitro and in vivo data of elemene suggest that it may be useful in the treatment of certain cancers that develop drug resistance. With the development of research, its molecular mechanism has become increasingly clear and plays an important role in overcoming drug resistance, mainly through inhibiting ABCB1 transporter efflux, reducing exosome delivery, regulating related genes, promoting apoptosis, regulating the cell cycle, inducing autophagy, and inhibiting stemness. In summary, elemene reverses drug resistance through multiple pathways and multiple targets, the mechanism of action is further clarified, providing valuable information for the study of tumor drug resistance, and it is also of great significance in the clinical study of reversing tumor multidrug resistance. Future controlled clinical trials are needed to evaluate the potential role of elemene in reversing drug-resistant cancers.

## Figures and Tables

**Figure 1 molecules-26-05792-f001:**
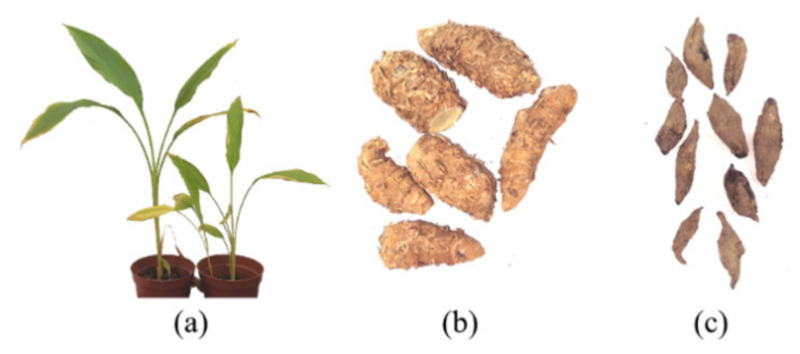
(**a**) The plant of *Curcuma wenyujin*; (**b**) *Curcuma zedoaria*, the dried rhizome of *Curcuma wenyujin*; (**c**) *Curcumae radix,* the dried tuberous root of *Curcuma wenyujin*.

**Figure 2 molecules-26-05792-f002:**
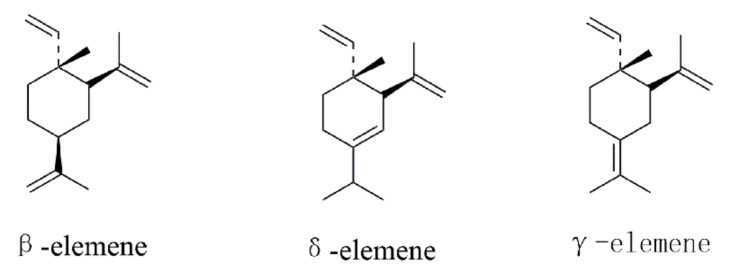
The structural formula of elemene.

**Figure 3 molecules-26-05792-f003:**
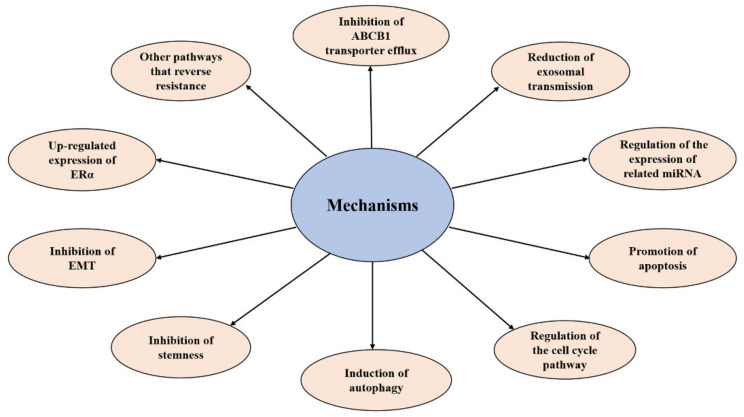
Classification of multidrug resistance mechanism of tumor cells reversed by elemene.

**Figure 4 molecules-26-05792-f004:**
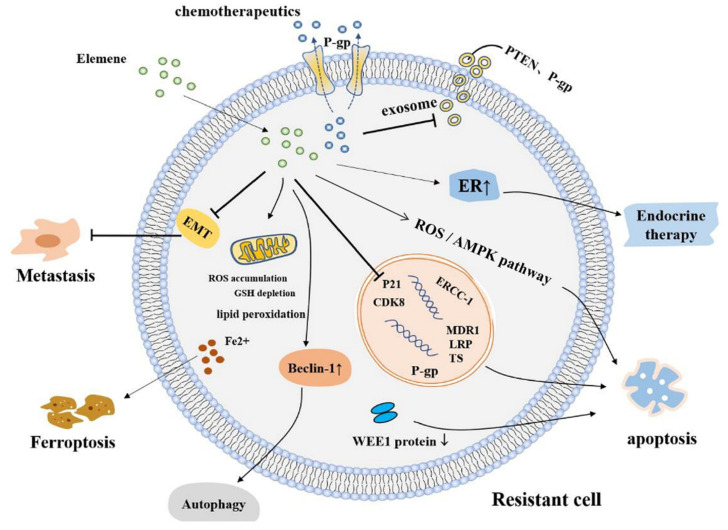
Classification of multidrug resistance mechanism of tumor cells reversed by elemene. The mechanisms of elemene reversing multidrug resistance include the induction of apoptosis, induction of iron death, induction of autophagy in tumor cells, inhibition of tumor cell migration, inhibition of the expression of P-gp efflux pump, inhibition of the expression of some MDR-related miRNAs in exosomes, and so on. The symble ↑ indicates upregulated expression. The symble ↓ indicates downregulated expression.

**Figure 5 molecules-26-05792-f005:**
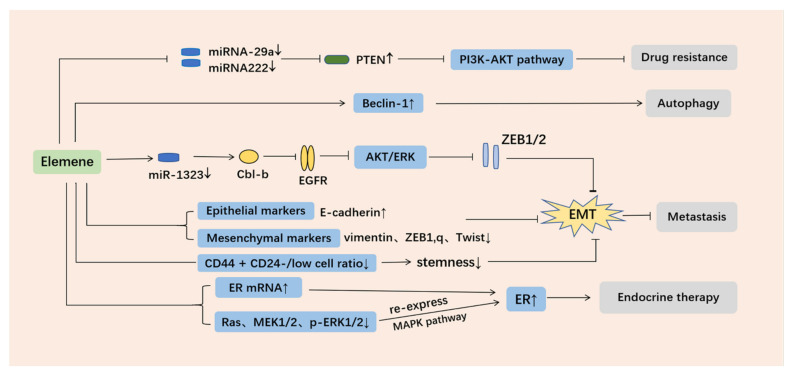
Mechanisms of elemene reversing multidrug resistance in tumor cells. Elemene reverses drug resistance by inducing autophagy and apoptosis, inhibiting stemness, inhibiting EMT, inhibiting tumor metastasis, and promoting endocrine therapy. Elemene induces autophagy by promoting Beclin-1. Elemene reverses drug resistance by inhibiting the PI3K–AKT signaling pathway and increasing *PTEN* gene expression. Elemene inhibits stemness by decreasing the CD44^+^CD24^−/low^ cell ratio and inhibits EMT by upregulating the expression of epithelial markers and downregulating the expression of mesenchymal markers. Elemene promotes endocrine therapy by increasing the expression of ERα, increasing the expression of mRNA, and re-expressing ERα through the MAPK pathway. The symble ↑ indicates upregulated expression. The symble ↓ indicates downregulated expression.

**Figure 6 molecules-26-05792-f006:**
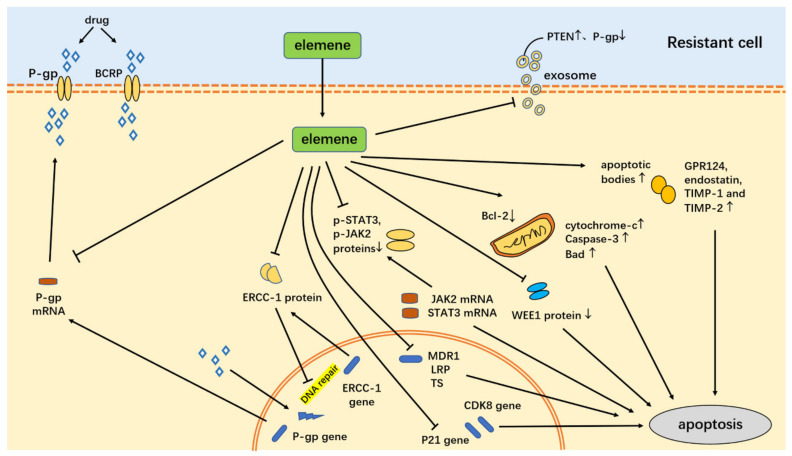
Pathways of elemene reversing drug resistance through the apoptotic pathway. The symble ↑ indicates upregulated expression. The symble ↓ indicates downregulated expression.

**Figure 7 molecules-26-05792-f007:**
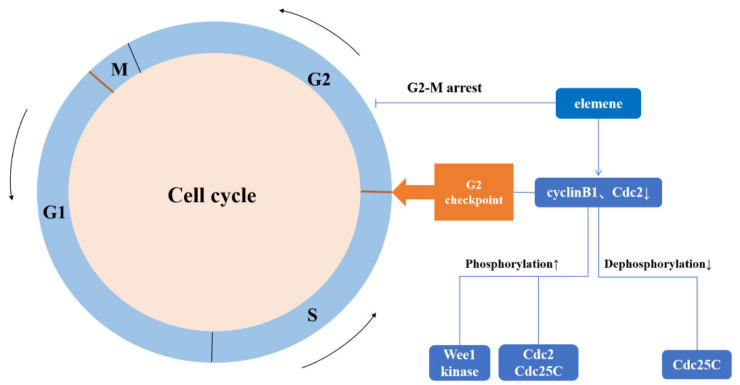
Pathways of elemene reversing drug resistance through regulation of the cell cycle pathway. β-Elemene blocks the G2–M cell cycle by regulating the cell cycle G2 checkpoint. Cyclin B1 and Cdc2, the cell cycle G2 checkpoints, are inactivated by an increase in Wee1 kinase phosphorylation, decreased Cdc25C dephosphorylation, and increased Cdc2 and Cdc25C phosphorylation. The symble ↑ indicates upregulated expression. The symble ↓ indicates downregulated expression.

**Figure 8 molecules-26-05792-f008:**
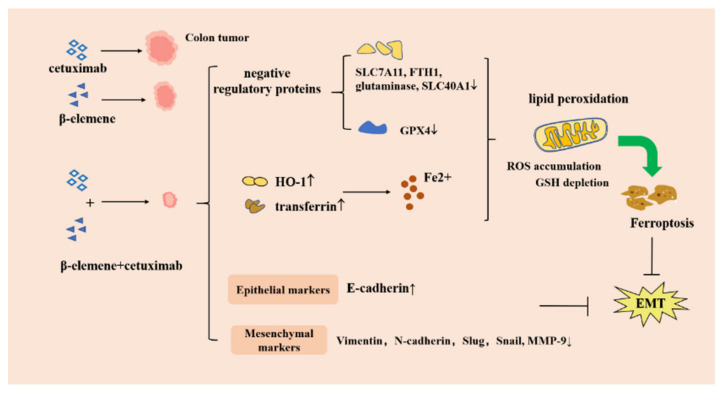
Mechanism of β-elemene inhibition of epithelial–mesenchymal transformation. The symble ↑ indicates upregulated expression. The symble ↓ indicates downregulated expression.

**Table 1 molecules-26-05792-t001:** Mechanisms of elemene reversing multidrug resistance in cells.

	Reversal of MDR Pathways	Mechanisms	Resistant Drug	Cells	Cancer	References
1	Inhibition of ABCB1 transporter efflux	BCRP and P-gp	MTO	K562/DOX	Leukemia	[41]
2	Reduce exosomal transmission	PTEN, P-gp	ADR, DOC	MCF-7/ADR, MCF-7/DOC cells	Breast cancer	[55]
		Inhibition of miR-1323/Cbl-b/EGFR pathway	-	-	Gastric carcinoma	[57]
		Inhibition of PI3K–AKT signaling pathway (*PTEN*)	ADR, DOC	MCF-7/Adr, MCF-7/DOC	Breast cancer	[37]
4	Promote apoptosis	Mitochondrial apoptosis pathway	DDP	A549/DDP cells	LA	[58]
		Upregulation of Caspase-3 protein	ADR	SGC7901/Adr cells	Gastric carcinoma	[21]
		Decreasing the expression of resistance genes *MDR1*, *LRP*, and *TS*	PTX	U-2OS cells	Bone neoplasms	[60]
		Suppressing the STAT3 signaling pathway	cisplatin (DDP)	YD-38 cells	GSCC	[61]
		Inhibition of *CDK8* gene	ADR	A549	Lung cancer	[75]
		Inhibition of *P21* gene expression	ADR	A549	Lung cancer	[76]
		Destroy DNA repair activity	DDP	A2780/CP70 cells	Ovarian cancer	[62]
		Inhibition of WEE1 expression	VCR	SW-480 cells	Colon cancer	[77]
5	Block the cell cycle pathway	Regulate the G2 checkpoint of the cell cycle and blocking the G2–M phase	DDP	A2780/CP	Ovarian cancer	[63]
6	Induce autophagy	Induction of Beclin-1 expression	DDP	SPC-A-1/DDP cells	LA	[64]
7	Inhibit the stemness	EZH2↓	GEF	NSCLC cells	NSCLC	[78,79]
		Inhibit the stemness, BCRP, P-gp	ADR	MCF-7/ADM cells	Breast cancer	[67]
8	Inhibit EMT	Block G0/G1 phase, mesenchymal markers (vimentin, N-cadherin, Slug, Snail, and MMP-9) ↓, epithelial marker E-cadherin	Cetuximab	KRAS mutated CRC cells	mCRC	[69]
9	Upregulate the expression of ERα	Upregulation of ERα mRNA and re-expression of ERα through the MAPK pathway	TAM	MCF-7/TAM cells	Breast cancer	[70]
		Inhibiting the c-Met signaling pathway	GEF	PC-9 cells	LA	[80]
		Upregulation of E3 ubiquitin ligase, c-Cbl, and Cbl-b	DNR/ADR	K562/DNR and SGC7901/ADR cells	Leukemia/gastric carcinoma	[72]
		Inhibition of the ERK1/2 -Bcl-2/Survivin pathway	TMZ	U251	GBM	[73]
		Inhibition of PI3K/Akt/mTOR signaling pathway	GEF	NCI-H460	LA	[81]

Abbreviation: BCRP, breast cancer resistance protein; P-gp, permeable glycoprotein; PTEN, Phosphatase And Tensin Homolog; Cbl, Casitas B-lineage lymphoma; EGFR, Epidermal Growth Factor Receptor; PI3K, Phosphatidylinositol 3 kinase; MDR1, multidrug resistance gene1; LRP, Lung Resistance protein; TS, tumor suppressor; STAT3, Signal Transducer and Activator of Transcription; CDK, Cyclin-Dependent Kinase; EZH2, enhancer of zeste homolog 2; MMP, matrix metallopeptidase; ER, estrogen receptor; MAPK, mitogen-activated protein kinase; ERK1/2, extracellular-regulated kinase1/2; Bcl, B-cell lymphoma; AKT, Protein Kinase B; mTOR, mammalian target of rapamycin; MTO, Mitoxantrone; ADR, Adriamycin; 5-FU, 5-Fluorouracil; DDP, Cisplatin; PTX, Paclitaxel; VCR, Vincristine; DNR, Daunorubicin; TMZ, Temozolomide; GEF, Gefitinib; GSCC, Gingival squamous cell carcinoma; NSCLC, Non-small cell lung cancer; GBM, Glioblastoma multiforme; mCRC, metastatic colorectal cancer; LA, Lung adenocarcinoma. The symble ↓ indicates downregu-lated expression.

**Table 2 molecules-26-05792-t002:** Mechanisms of inhibition of ABC transporter efflux by β-elemene.

Therapeutic Drugs	Cells	Mechanisms
β-elemene (100 µM) + colchicine, vinblastine, paclitaxel	Overexpressing ABCB1 transporter KB-C2 cells	Inhibition of ABCB1 transporter efflux (P-gp)
β-elemene (100 µM) + mitoxantrone	BCRP-overexpressing NCI-H460/MX20 cells	Partially (~50%) increased the sensitivity; Inhibition of ABCB1 transporter efflux (BCRP)
MTO/βE-SLNs	K562/DOX cells	Increased cellular uptake and blockage of intracellular ATP production and P-gp efflux by βE
β-elemene + DOX	MCF-7/DOX fluc cells	Downregulating P-gp, MRP, and BCRP gene and protein expression
β-elemene + GEF	Gefitinib-resistant PC-9/GR fluc cells	Inhibit the efflux function of ABC drug-resistant proteins

## Data Availability

Not applicable.

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
