# Peer review of "Recent Advances in Understanding the Mechanisms of Elemene in Reversing Drug Resistance in Tumor Cells: A Review"

_molecules, 2021, doi:10.3390/molecules26195792_

Round 1

Reviewer 1 Report

Elemene, an extract of the natural drug Curcuma wenyujin reverses drug resistance and sensitize cancer cells to chemotherapy by inhibiting the efflux of ATP binding cassette subfamily B member 1(ABCB1) transporter, reducing the transmission of exosomes, inducing apoptosis and autophagy.  Moreover, it regulates the expression of important genes and proteins in various signaling pathways, blocking the cell cycle, inhibiting stemness, epithelial-mesenchymal transition. This review article describes the mechanisms of elemene's reversal of drug resistance in detail. The manuscript is comprehensive and well-organized as well as each section is thoroughly discussed.

Author Response

Special thanks to you for your good comments and acknowledgement. Thank you very much for your patience in reading my manuscript and for acknowledging my manuscript ideas.

Reviewer 2 Report

In this review manuscript,  Tan et al summarized the mechanism underlying reversing drug resistance of  Elemene, a natural compound from plants. It is interesting. The following comments may be helpful for enhancing the significance of the compound.

  • It seems that the Elemene is more like dietary supplements, rather than a clinic anti-tumor drug or potential drug.
  • How does the compound, Elemene work on its targets, by direct binding? If so, what is the specific target of the compound?
  • The compound have effect on almost all oncogenic pathways so far reported, why and how?
  • This review does not emphasize the in vivo animal data for the anti-tumor effect and reversing drug resistance of the compound in the clinic, either single compound or combination.
  • The review did not emphasize, possibly it lacks the clinical trial data for the anti-tumor effect and reversing drug resistance of the compound in the clinic, either single compound or combination.

Author Response

Special thanks to you for your good comments and suggestions. Those comments are all valuable and very helpful for revising and improving our paper, as well as the important guiding significance to our researches. We have studied comments carefully and have made correction which we hope meet with approval. Please see the attachment.

Reviewer 3 Report

General comment:

This is a review article that Tan et al tried to summarize the impact of Elemene on cancer chemoagent resistance, and its molecular mechanism on the reversal process of multidrug resistance. This is a well written and organized review article in the perspective of elemene. Since this journal particular interest to the molecule interactions of biological activities, the pharmaceutical, pharmacological, and toxicological concern can be ignored in this case, although it’s critical to discuss a single compound.

Detailed Comment

  1. Line 147: microrna-191à miRNA-191
  2. Reference 49 is nothing direct linking to TS. Also, what tumor suppressor did author refer to?
  3. There’s no discussion regarding the differential effect of elemene isomers (beta, delta, and gamma) on the reversal process of multidrug resistance.
  4. The conclusion section was a paragraph condense from abstract and introduction with minor revision. Reviewer suggest that authors can shorten this paragraph in another way of interpretation. And strongly recommend to add a paragraph to discuss the possibility of developing clinical useful agent.
  5. There are many difficulties of translating traditional herb medicine to clinical useful agent. One of them is to identify active component. Another one is to identify primary target molecule/protein of the active component that trigger biological consequences. Since elemene is single compound with multiple impact on cellular signals, it’s important to discuss possible target of elemene that responsible for those wide-spread downstreams. This will greatly provide insight to the biological significance of elemene in cancer. Besides, it would shed light to next step of translating elemene from functional herb ingredient into a real drug. Therefore, reviewer strongly urge author to consider put this discussion paragraph in the article.

Author Response

(The authors gave the same response as above.)

Round 2

Reviewer 2 Report

In the revised manuscript, the authors answered most of my questions and concerns. It is a benefit that natural compounds have fewer side effects, however, they may lack the specific pharmaceutic targets.  

Several review papers have been published about elemene from the same laboratory; also the reviews concerning the drug resistance in cancer have been published (Oncol  Rep,  2014;31(5):2131-8; Chinese J Cancer, 2015).

(1) Elemene is a chemotherapeutic compound, the major mechanism should be cytotoxicity, then some of the summarized mechanisms are responsible for cytotoxicity.    It is better to focus more on the mechanisms underlying drug resistance by summarizing the reports from in vitro cell level, in vivo animal level, and clinic therapy level. 

 (2) If the direct effect of Elemene is to bind the target protein, then the effect of Elemene on small RNAs and other targets may not be direct. It is better to discuss more. Also, it is better to have a comment about the more study needed and what improvement is needed.

(3) does Elemone have an effect on immune function? if so, it is related to drug resistance, and how?

(4) The commercial marks should not be shown in Figure 1

Author Response

Thank you for reading my revision and giving further valuable comments and suggestions. These comments have important guiding significance for our research. We have carefully studied the comments and corrected them in the hope of approval. The major revisions in the manuscript are marked. Please see the attachment.

Reviewer 3 Report

Nil.

Author Response

Thank you for carefully reading my manuscript revision and your recognition of my revision. Thank you again for your comments and suggestions during the first review to make my article more perfect after revision. Best wishes to you.